# The effect of monetary incentive on survey response for vulnerable children and youths: A randomized controlled trial

Jan Hyld Pejtersen ⓘ *

VIVE–The Danish Center for Social Science Research, Copenhagen, Denmark

* jhp@vive.dk

## Abstract

### Aim

In surveys non-responders may introduce bias and lower the validity of the studies. Ways to increase response rates are therefore important. The purpose of the study was to investigate if an unconditional monetary incentive can increase the response rate for vulnerable children and youths in a postal questionnaire survey.

### Methods

The study was designed as a randomized controlled trial. The study population consisted of 262 children and youth who participated in an established intervention study aimed at creating networks for different groups of vulnerable children and youths. The mean age of the participants was 16.7 years (range 11–28) and 67.9% were female. The questionnaire was adapted to three different age groups and covered different aspects of the participants' life situation, including the dimensions from the Strengths and Difficulties Questionnaire (SDQ). In the follow-up survey, participants were randomly allocated to two groups that either received a €15 voucher for a supermarket together with the questionnaire or only received the questionnaire. We used Poisson regression to estimate the differences in response rate (Rate Ratio RR) between the intervention group and the control group.

### Results

The response rate was 75.5% in the intervention group and 42.9% in the control group. The response rate in the intervention group was significantly higher than in the control group when adjusting for age and gender (Rate Ratio, RR 1.73; 95% CI 1.38–2.17). We did not find any significant differences in scales scores between the two groups for the five scales of the SDQ. In stratified analyses, we found the effect of the incentive to be higher for males (RR 2.81; 95% CI 1.61–4.91) than for females (1.43; 95% CI 1.12–1.84).

### Conclusions

Monetary incentives can increase the response rate for vulnerable children and youths in surveys.

**Data Availability Statement:** All relevant data are within the paper and its Supporting Information files.

**Funding:** The data was collected as part of a study funded by Danish National Board of Social

Services, project (in Danish): Lige muligheder – Udsatte børn og unge. The funder had not role in any aspect of the study.

**Competing interests:** The authors have declared that no competing interests exist.

## Trial registration

The trial was retrospectively registered at ClinicalTrials.gov Identifier: NCT01741675.

## Introduction

In questionnaire surveys non-responders reduce the effective sample size and may introduce bias, and this will lower the validity of the studies. The bias can occur when the non-responders are systematically different from those who participate in the survey. The likelihood of bias decreases with increasing response rate. Cohort studies show that response rates in epidemiological surveys have declined over time [1,2]. Moreover, Galea & Tracey suggest that this decline may be due to several factors, for example, a dramatic increase in the number of scientific surveys and especially surveys within telemarketing, and a decrease in people's general willingness to participate in community activities [1]. Consequently, finding ways to improve response rates has become increasingly important.

A Cochrane review by Edwards et al. identified trials that evaluated different ways of increasing response rates [3]. The trials evaluated more than 110 different strategies for increasing response to postal questionnaires. Different types of incentives were examined in most of the trials, and it was seen that not only did monetary incentives significantly increase the response rate to postal questionnaires (odds ratio (OR) 1.87), they also had a greater effect than non-monetary incentives (OR 1.62). Moreover, the response rate increased when the incentive was given unconditionally together with the questionnaire compared with when it was given under the condition that the respondents return their questionnaires (OR 1.61). Newer studies are in line with the review by Edwards et al. [4–7].

Previous studies of the effect of incentives on response rates in surveys have been conducted within various groups, for example, adults in the general population, employees, residents, households, businesses, patients, health personnel, consumers and students [3–7]. However, to our knowledge, none of the studies concerned vulnerable children and youths.

While much research has focused on studying the effect of incentives on response rates in surveys, little attention has been given to studying the effect of incentives on the response distribution [8]. That is, whether the incentive changes the response in the treatment group compared to the control group. Incentives can affect the response distribution indirectly by affecting the composition of demographic variables, such as gender and age of the study sample, or directly by affecting the attitude of the respondents [8]. There may be a risk that respondents who receive an incentive are more likely to provide positive answers than the control group who do not receive an incentive.

Even though advances in communication technology continue to alter the way we perform surveys, postal questionnaires are still commonly used in surveys within health and social sciences. There may be several reasons for this. First, despite the increasing popularity of web or online surveys, postal questionnaire surveys still have a higher response rate than web surveys [9–12]. Having said that, a few studies have shown similar response rates for the two different modes [13–15]. Second, surveys based on postal questionnaires are relatively easy to administer, do not require online access and may have less bias than web surveys, because responses in web surveys may depend on the age and level of education of respondents [12,16,17]. This is why a postal questionnaire survey may be the preferred method within social sciences when dealing with vulnerable groups.

The purpose of the study was to investigate if an unconditional monetary incentive can increase the response rate for vulnerable children and youths in a postal questionnaire survey.

We hypothesized that participants who were randomly allocated to receive a supermarket voucher (€15) together with the questionnaire would have higher response rates than participants in a control group, who only received the questionnaire. We also investigated if the monetary incentive influenced the response distribution in the postal survey.

## Materials and methods

### Study design

The study was designed as a randomized controlled trial (RCT). To study the effect of incentives on survey response, participants were randomly allocated to receive a €15 supermarket voucher together with the questionnaire or to only receive the questionnaire. The trial was retrospectively registered at ClinicalTrials.gov Identifier: NCT01741675.

### Participants

Participants were taken from an evaluation study of a social initiative aimed at creating networks for different groups of vulnerable children and youths aged 8–23 years [18]. Both the social initiative and the evaluation study were initiated by the Danish National Board of Social Services and were performed in ten municipalities in Denmark. The main purpose of the social initiative was to strengthen the social skills of the participants as well as their ability to create social relations. The children and youth all came from families with severe social problems. Moreover, they were vulnerable and lonely, because they had an inadequate or severely restricted social network or were excluded from their peer group. Participants came from three main groups: children in foster care or former foster care youths; children from vulnerable families; and lonely children and youths. The children from vulnerable families included children from families with a mentally ill or physically disabled parent or sibling, children of parents with alcohol abuse problems and children raised in violent families. The group characterized as lonely children and youths was a group of children and youths that did not fit into the other two groups.

In the evaluation study, written informed consent was obtained for all participants. Approximately 70% of the children and youths who participated in the social initiative gave consent to participate in the evaluation study. For participants under 16 years old, written informed consent was obtained from a parent, and for all other participants, consent was given by the participants themselves. The RCT study was submitted to the National Committee on Health Research Ethics (reference number VEK H-1-2012-FDP-79), who declared that questionnaire surveys do not need approval by the ethics committee in Denmark. The notification exemption for studies that only include questionnaire surveys is described in part 4, section 14(2) of the Danish Act on Research Ethics Review of Health Research Projects [19].

Participants who gave informed consent to take part in the evaluation study of the social initiative, and thereby agreed to fill out questionnaires and report their unique personal identification number, were eligible to take part in the present RCT study. We included participants regardless of whether they had responded to the questionnaires in the original study. In total, 274 participants were eligible for the study. Based on the participants' unique personal identification number (CPR number in the Danish Civil Registration System), the Central Office of Civil Registration provided us with addresses of the participants. Participants who were not available in the Civil Register because they were dead, had immigrated or had requested survey exemption were excluded from the study. Participants who reported an incomplete identification number were also excluded. Furthermore, participants who had changed address and had not yet reported their new address to the local authorities were also excluded from the study.

## Survey

The questionnaire survey in the present study was a follow-up survey of the original study and was performed in four waves. The first wave was performed in November 2012 and included participants who had left the original study 1–2½ years before the follow-up. The second, third and fourth waves were performed in May 2013, November 2013 and May 2014, respectively, 1–1½ years after the participants had left the original study.

## Randomization

The randomization was performed within each wave of the survey. The participants were given a unique identification number in the original study. Within each wave of the study, participants were sorted according to their identification number and were assigned a random number between 0 and 1. Participants assigned with a number lower than 0.5 were allocated to the control group; participants assigned with a number greater than 0.5 were allocated to the intervention group. We used the Rand ('uniform') function in SAS to create the randomization sequence.

## Intervention

The intervention group received a postal questionnaire together with a voucher worth €15 for the largest supermarket chain in Denmark. The control group only received a questionnaire. The questionnaire was sent to both the intervention group and the control group together with a cover letter and a stamped return envelope. The addresses on the envelopes were hand-written. After three weeks, non-responders in both groups were sent a reminder together with a new copy of the questionnaire.

After 13 weeks, the study period was over and the participants in the control group received a voucher similar to the one given to the intervention group. This was done for ethical reasons to ensure both groups were treated equally. Responses received after the study period were not included in the analyses.

## Questionnaire

As the age of the study population ranges from 8 to 23 years, the questionnaire was made in three versions covering three age groups: 8–12 year olds, 13–16 yea -olds and 17–23 year olds. The questionnaires for the 8–12 year olds and 13–16 year olds included 91 items and the questionnaire for the 17–23 year olds included 90 items. In total, 78 of the items were the same for the three age groups, 13 items where the same for two age groups and 12 items were unique for one age group.

The questionnaire covered the following areas regarding the participants' life situation: family and housing; education and training; sport and leisure time; relation to friends; drug use; and strengths and difficulties. The questionnaire also included questions that evaluated the intervention in the original study. These questions were taken from three Danish longitudinal surveys of children and youths [20,21]. The Danish version of the Strengths and Difficulties Questionnaire (SDQ) was included in the questionnaires for all three age groups [22].

## Outcome measure

The primary outcome measure was the questionnaire response rate that was defined as the proportion of questionnaires returned by participants. The secondary outcome was scores on the five multi-item scales in the strengths and difficulties questionnaire.

## Statistical analysis

In the primary analysis we estimated the rate ratio (relative risk, RR) to assess whether there was a difference between group allocation and questionnaire response rate. We used Poisson regression with robust error variance [23] and adjusted for age and gender in the analysis. The analysis was performed using PROC GENMOD in SAS.

In the secondary analyses we tested whether mean scores on the five SDQ scales were different between group allocations. We adjusted for the covariates age and gender as in the primary analysis. We used GLM in SAS. The significance level was set at 5% in all analyses.

In post-hoc analyses we calculated the rate ratio for the response rate when stratified by gender. Furthermore, as a robustness test of the primary analysis, we calculated the rate ratio where we adjusted for response history. Response history was defined as whether the respondent had responded to the questionnaire in the original study or not. In a final robustness test, we calculated the rate ratio where we adjusted for the group that characterized the children and youths.

The protocol of the trial was registered at ClinicalTrials.gov Identifier: NCT01741675.

## Results

In total, 274 participants were eligible for the study (Fig 1). However, we excluded seven participants from the study before the randomization. Three participants had emigrated at the time of the follow-up, two had an error in their personal identification number (CPR), and two had no known address. Of the 267 participants who were randomized, 146 were allocated to the intervention group and 121 were allocated to the control group. We sent questionnaires to the 267 participants in the study population. However, we had to exclude three participants in the invention group and two participants in the control group because the letter could not be delivered to their addresses that were obtained from the CPR register. Consequently, a total of 262 participants were included in the primary analysis.

The characteristics of the study population and the respondents are shown in Table 1. The original study aimed at children and youths aged between 8 and 23 years, but participants older than 23 years of age were also included. At the follow-up that was performed 1–2½ years after participants were included in the original study, the age of the study population ranged between 11 and 28.

The original study was organised into twelve projects within the ten municipalities. The projects all worked at creating networks for the participating children and youths. The population consisted of 67 children in foster care or former foster care youths, 77 children and youths from vulnerable families and 60 participants characterized as lonely children and youths, Table 1. A total of 58 participants came from projects that included participants from all target groups, Table 1.

A total of 159 participants returned the questionnaire, resulting in an overall response rate of 60.7% for the study. The response rate in the intervention group was 75.5%, whereas the response rate in the control group was 42.9%, Table 2.

The response rate in the intervention group was significantly higher than in the control group when adjusting for age and gender (RR 1.73; 95% CI 1.38–2.17), Table 3.

In the secondary analyses we tested whether scores for the five SDQ scales differed between the intervention group and the control group. The mean scores for the five scales in the SDQ are given in Table 4. When comparing the five scale scores, we found no significant difference between the scores for two groups when adjusting for age and gender. We found similar results in analyses where we did not adjust for gender and age.

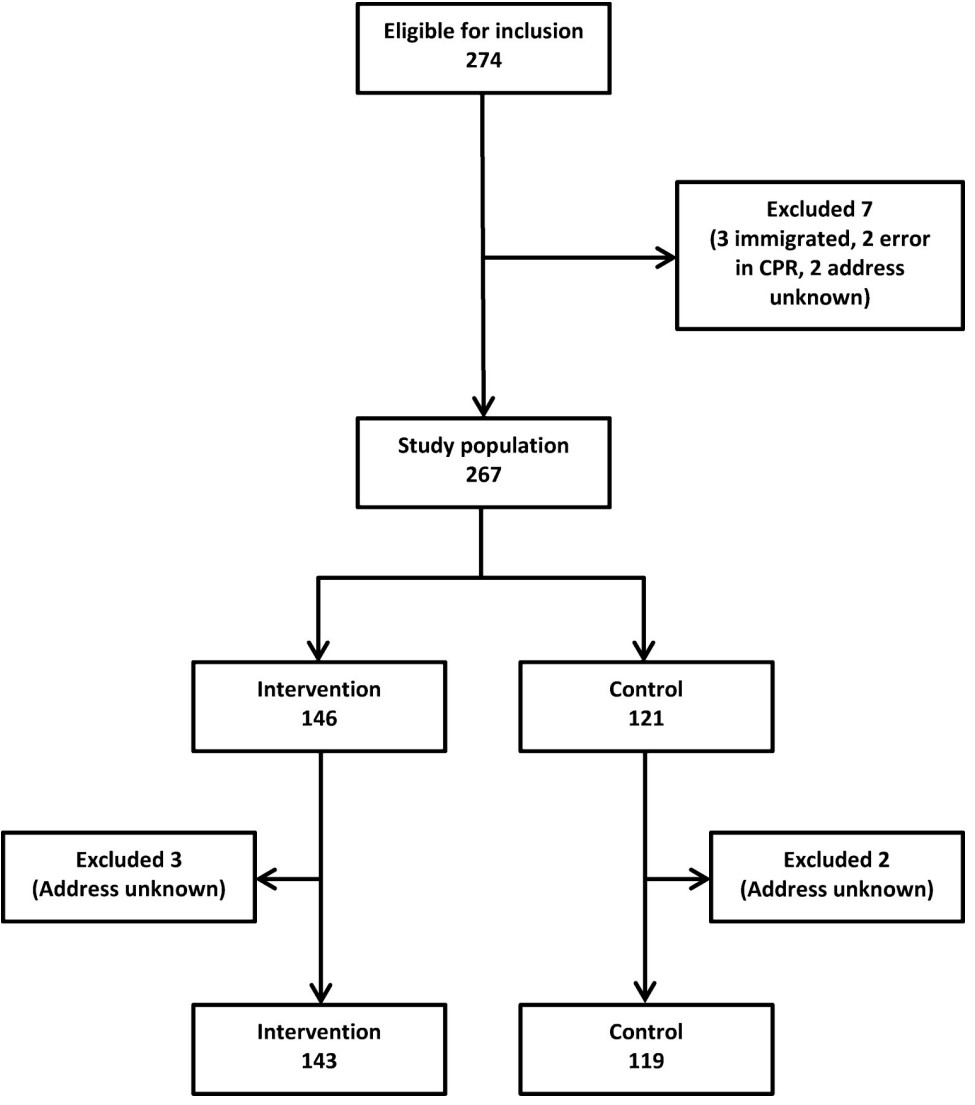

**Fig 1. Flow diagram for participants included in the trial.**

In Table 1, the characteristics of participants who answered the questionnaire are given. The percentage of females who answered the questionnaire was remarkably higher in the control group than in the intervention group and vice versa for men. We therefore performed a post hoc analysis on the total sample in which we analysed the response rate and allocation stratified by gender.

In the stratified analyses, the response rate was significantly higher for both males and females in the intervention group compared to in the control group. For males, the rate ratio was 2.81; 95% CI 1.61–4.91 (p = 0.0003). For females, the rate ratio was 1.43; 95% CI 1.12–1.84 (p = 0.0048).

We tested if the response history in the original study could alter the conclusion of the primary analysis. In total, 58% of the participants had answered at least one questionnaire in the original study; 42% were non-responders. Adjusting for response history did not change the rate ratio much (RR 1.71; 95% CI 1.36–2.14), and response history was insignificant in the analysis (p = 0.07).

**Table 1. Characteristics of the study population and respondents at follow-up.**

|  | Intervention | Control | Total |
|---|---|---|---|
| Participants |  |  |  |
| Number | 143 | 119 | 262 |
| Female, number | 96 | 82 | 178 |
| Female, % | 67.1 | 68.9 | 67.9 |
| Male, number | 47 | 37 | 84 |
| Male, % | 32.9 | 31.1 | 32.1 |
| Age, mean (standard deviation) | 16.3 (3.5) | 17.1 (3.6) | 16.7 (3.6) |
| Age range | 11–26 | 11–28 | 11–28 |
| Children in foster care or former foster care youths | 35 (52.2) | 32 (47.8) | 67 |
| Children from vulnerable families | 43 (55.8) | 34 (44.2) | 77 |
| Lonely children and youths | 34 (56.7) | 26 (43.3) | 60 |
| A mixture of all 3 target groups | 31 (53.5) | 27 (46.6) | 58 |
| Respondents |  |  |  |
| Number | 108 | 51 | 159 |
| Female, number | 69 | 41 | 110 |
| Female, % | 63.9 | 80.4 | 69.2 |
| Male, number | 39 | 10 | 49 |
| Male, % | 36.1 | 19.6 | 30.8 |
| Age, mean (standard deviation) | 16.2 (3.5) | 16.5 (3.0) | 16.3 (3.4) |
| Age range, years | 11–26 | 12–23 | 11–26 |
| Children in foster care or former foster care youths | 25 (65.8) | 13 (34.2) | 38 |
| Children from vulnerable families | 34 (66.7) | 17 (33.3) | 51 |
| Lonely children and youths | 28 (73.7) | 10 (26.3) | 38 |
| A mixture of all 3 target groups | 21 (65.6) | 11 (34.8) | 32 |

In a final analysis, we adjusted the primary analysis for the group variable that characterized the children and youths. This did not change the rate ratio (RR 1.73 CI 1.38–2.17). The group variable was insignificant in the analysis.

## Discussion

Our results show that monetary incentives can significantly increase the response rate for vulnerable children and youths in postal surveys. Despite the differences in response rate between the intervention and control group, we did not find any significant difference in SDQ scales scores between the two groups.

To our knowledge, this is one of the first studies on the effect of monetary incentives on survey response for vulnerable children and youths. The children and youths who unconditionally received the incentive together with the questionnaire were 1.73 times more likely to respond to the questionnaire compared to the children and youths who only received the questionnaire. In other words, the response rate in the intervention group was 73% higher than in the control group.

**Table 2. Response rate for the intervention and control group.**

|  | Intervention | Control |
|---|---|---|
| Response rate (%) | 75.5 | 42.9 |
| Responders (number) | 108 | 51 |
| Total (number) | 143 | 119 |

**Table 3. Difference in response rate for the intervention and control group (rate ratio, relative risk).** The model was adjusted for age and gender*.

| Allocation | Rate ratio RR | 95% confidence interval | P-value |
|---|---|---|---|
| Control | 1 | | |
| Intervention | 1.73 | 1.38–2.17 | <0.0001 |

*Gender and age were not significant in the model

In our analysis we calculated the rate ratio because, in contrast to the odds ratio, it can be interpreted directly. To compare our results with the results presented in the Cochrane review [3], we estimated the odds ratio based on the rate ratio and the incidences in our study [24]. The rate ratio of 1.73 corresponded to an odd ratio of 4.0. However, the results from the Cochrane review cannot be compared directly to the results of the present study as we must combine two odds ratios into one. In the Cochrane review, the effect size for monetary incentive vs no incentive was OR = 1.87, and the effect size for unconditional vs conditional monetary incentive was OR = 1.61. In the studies that compared monetary incentive vs no incentive, the incentive was based on a combination of conditional incentive and unconditional incentive. In the Cochrane review, there was no direct comparison of the treatment unconditional monetary incentive vs no incentive. In accordance with Bucher et al. [25], if we regard the meta-analysis of the treatment monetary incentive vs no-incentive presented in the Cochrane review as a treatment of conditional monetary incentive vs no-incentive, we can combine the two indirect comparisons into one direct comparison [25,26]. The effect size for unconditional incentive vs no incentive was then calculated from the two separate effect sizes OR = 1.61/(1/1.87) = 3.0. The effect size of OR = 4.0 (RR = 1.73) in our study was larger than the effect size of OR = 3.0 in the Cochrane review. This emphasizes that the effect size in our study was a remarkably large effect size compared to that found in previous studies [3]. Furthermore, the present study comprised vulnerable children and youths for whom a low or moderate response rate would be expected, whereas the studies in the Cochrane Review comprised adults and less vulnerable groups.

The strength of the study was that it was designed as an RCT study. The random allocation ensured that there were no systematic differences between the intervention group and the control group, and therefore the association found between the intervention and the outcome has a high probability of being a causal relationship. Furthermore, before the study was performed, we registered a study protocol in which we outlined the planned analyses.

The study had some limitations. The study population was not homogenous as it comprised different groups of vulnerable children and youths. However, despite this lack of homogeneity, all participants shared the same problem of having limited social relations and, consequently

**Table 4. Mean and standard deviation for the five scales in the SDQ.** The regression models are adjusted for age and gender.

| | Intervention N = 105* | | Control N = 51 | | P-value |
|---|---|---|---|---|---|
| | Mean | Std | Mean | Std | |
| Emotional symptoms | 4.0 | 2.9 | 4.7 | 2.5 | 0.4552 |
| Conduct problems | 2.0 | 1.7 | 2.0 | 1.6 | 0.8216 |
| Hyperactivity/Inattention | 4.4 | 2.5 | 4.0 | 2.6 | 0.3399 |
| Peer relationship problems | 2.6 | 1.9 | 3.3 | 2.4 | 0.0721 |
| Prosocial behaviour | 7.7 | 2.1 | 7.5 | 1.9 | 0.4643 |

*108 respondents returned and answered the questionnaire. However, three respondents did not answer the questions regarding the SDQ items.

they were lonely or at risk of being lonely. Furthermore, in the analysis where we adjusted the primary analysis for the group variable that characterized the children and youths, we did not see any change in the rate ratio.Reaching socially disadvantaged or vulnerable groups in surveys can be challenging [27]. Nevertheless, to our knowledge, there is no evidence that different vulnerable groups differ with regard to their willingness to participate in surveys.

The study population had a quite large age span from 10 to 28 years (mean age 16.7) and was not evenly distributed with respect to gender (67.9% of the participants were female). Because demographic variables such as gender and age may influence response rates in surveys, we decided to adjust for these variables in the analyses. The primary analysis showed no statistically significant difference for gender and age in the model (see Table 3), and therefore the found result was not biased by age and gender. Similarly, in the secondary analyses of the effect of the incentive on the response distribution, the results were independent of whether we adjusted for gender and age.

We used simple randomization within each of the four waves of the study. As a result of this procedure, 146 participants were allocated to the intervention group and 121 participants were allocated to the control group. In a small sample like this, one could argue that we should have ensured equal allocation to the two groups by using block randomization. However, even with the simple randomization procedure we used, the difference in the two groups are the result of chance rather than bias.

In the secondary analyses, we did not find any significant difference in SDQ scales scores between the intervention group and the control group. However, the results should be interpreted with care as we had limited power for the analyses. These results should be replicated for a larger sample.

Our stratified analysis showed that the effect of the incentive was larger for males (RR = 2.81) than for females (RR = 1.43). The gender distribution of the respondents in the intervention group (63.9% female) was close to the gender distribution of the total sample (67.9%). In comparison, the percentage of females who responded in the control group was 80.4%. The monetary incentive reduced or practically levelled out the gender bias that was seen in the control group. Thus, a monetary incentive may reduce any gender bias.

The present study was performed on a group of participants who were already participants in an ongoing study, and 58% of the participants had already answered at least one questionnaire in the original study. We do not know whether the effect of the incentive would have been smaller if the present survey had been the first contact to the participants. However, when we adjusted the primary analysis for response history, we did not see any change in the rate ratio.

We deviated from the pre-specified protocol with regard to two points. First, in the primary analysis we only specified that we would adjust for age, whereas in the secondary analyses we specified that we would adjust for age and gender. In reality, we adjusted for both age and gender in the primary analysis and secondary analyses. In the protocol, we specified that the follow-up period was 10 weeks. In reality, the follow-up period was 13 weeks, but only very few questionnaires were returned in the period from week 10 to 13.

The Strength and Difficulty Questionnaire was developed for children aged between 4 and 17. In the original intervention study, it was decided to use the SDQ for all participants even though the study aimed at children and youths aged between 8 and 23. Later it was further decided to expand the intervention to also include a few participants who were older than 23 years. However, in our analyses we did not find a significant effect of age when looking at differences in scale scores between the intervention group and the control group. In a recently published study, Brann et al. [28] have examined the psychrometric properties of SDQ for

young adults (age 18–25 years) in comparison with SDQ for adolescents (age 12–17 years). They found similar psychometric properties for the two versions of the SDQ.

The present study was performed with a postal questionnaire. We do not know if the results also are valid for web surveys for vulnerable children and youths. The Cochrane review found no effect of monetary incentives on the response rate in web surveys [3]. Recent reviews have found an effect of monetary incentives on the response rate in web studies, however, the effect is still smaller than for postal-based studies [29–31].

## Conclusions

We conclude that monetary incentives can increase the response rate in postal questionnaires for vulnerable children and youths. The monetary incentive had a larger effect for males than for females.

## Supporting information

**S1 Dataset. Dataset for the study.**
(XLSX)

## Author Contributions

**Conceptualization:** Jan Hyld Pejtersen.

**Data curation:** Jan Hyld Pejtersen.

**Formal analysis:** Jan Hyld Pejtersen.

**Funding acquisition:** Jan Hyld Pejtersen.

**Investigation:** Jan Hyld Pejtersen.

**Methodology:** Jan Hyld Pejtersen.

**Project administration:** Jan Hyld Pejtersen.

**Writing – original draft:** Jan Hyld Pejtersen.

**Writing – review & editing:** Jan Hyld Pejtersen.

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
