## [Decision Letter · Decision Letter 0]

9 Jan 2020

PONE-D-19-22509

The effect of monetary incentive on survey response for vulnerable children and youths: A randomized controlled trial

PLOS ONE

Dear Dr Jan Hyld Pejtersen,

Thank you for submitting your manuscript to PLOS ONE. After careful consideration, we feel that it has merit but does not fully meet PLOS ONE’s publication criteria as it currently stands. Therefore, we invite you to submit a revised version of the manuscript that addresses the points raised during the review process.

You will find the reviewers comments in this letter. They have raised issues about the methods and contribution to knowledge. Please could you address these concerns itemised in an table providing the comment, your response and the page and line number of any changes.

We would appreciate receiving your revised manuscript by 9th March 2020. To enhance the reproducibility of your results, we recommend that if applicable you deposit your laboratory protocols in protocols.io, where a protocol can be assigned its own identifier (DOI) such that it can be cited independently in the future. For instructions see: http://journals.plos.org/plosone/s/submission-guidelines#loc-laboratory-protocols

We look forward to receiving your revised manuscript.

Kind regards,

Dr Leica S. Claydon-Mueller

Academic Editor

PLOS ONE

Journal Requirements:

2.  Please include the ClinicalTrials.gov Identifier in your manuscript (NCT01741675)

3. Please clarify in your manuscript that your ethics committee specifically waived the need for ethics approval.

4. You indicated in your ethics statement that you obtained written informed consent from a parent for participants under 16 years old. Please clarify whether parental consent was also obtained from participants aged 16 and 17. If not, please clarify whether the research ethics committee specifically waived the need for their consent. Please add this information to your methods section.

Reviewers' comments:

Reviewer's Responses to Questions

**Comments to the Author**

1. Is the manuscript technically sound, and do the data support the conclusions?

Reviewer #1: Yes

Reviewer #2: Partly

2. Has the statistical analysis been performed appropriately and rigorously? 

Reviewer #1: Yes

Reviewer #2: I Don't Know

3. Have the authors made all data underlying the findings in their manuscript fully available?

Reviewer #1: Yes

Reviewer #2: No

4. Is the manuscript presented in an intelligible fashion and written in standard English?

Reviewer #1: Yes

Reviewer #2: No

5. Review Comments to the Author

Reviewer #1: Interesting and relevant topic however the use of postal questionnaire in health and social sciences are reducing day by day so not sure what this research will bring.The paper is generally well presented in a good literary style although there are a few comments which need to be addressed, for example:

1. The authors mentioned they used vulnerable children and youths in this research but its not clarified why they are considered as vulnerable groups.

2. I think it would be good update why non response rate among the children and youth is a major problem.

3. In introduction, I feel the reference number 1 was outdated and more recent one could be found and utilized.

4. Why children below 8 years old were included. Isn't it obvious they would not know the importance in participating in a research through postal survey.

5. line 126 to 128 " Participants with a number lower than 0.5 were allocated to the control group, participants with a number greater than 0.5 were allocated to the intervention group". This is not clear.

6. Very minimal statistical analysis performed. Table 1 and Table 5 can be merged and findings can be presented together.

I am still not clear how this research will contribute to the existing body of knowledge.

Reviewer #2: Note: I didn't see a link to the full data set within the PDF file I was provided. However, maybe the authors submitted this separately? i responded with a "no" to the question above (#3) as I don't see such a link, but it may exist outside of this PDF.

Note: I answered "I don't know" to question 2 as it was unclear from the manuscript how the randomization was completed. It is possible it was done correctly, but I don't know. The authors need to provide more information within the methods section of their paper.

---

Overall, the authors seem to be filling an important gap in the literature (conducting on RCT on the effectiveness of monetary incentives on an under-studied population). They also measure these effects on survey responses themselves. The use of an RCT is a reasonable way to meet their research objectives. However, there are a number of serious issues with the manuscript, including the extent to which these findings can be generalized to other populations, methodological inconsistencies, a lack of clarity regarding the make-up of the study sample, as well as various grammatical issues.

GRAMMAR

This paper could benefit tremendously from professional editing with attention to both grammar and diction. The language could also be more formal, and the conclusion in particular could be more fleshed out (it is only 2 sentences). Noting just a few issues on the first page or two:

(1) Noun/ verb disagreements: Different types of incentive(S); age group(S).

(2) The sentence starting on line 47 “Different types of incentive” is repetitive with the second half of that sentence.

(3) The sentence “The participating children and youths were vulnerable and lonely; because of they had inadequate or severely restricted social network or were excluded from their peer group” has multiple issues.

STUDY LOCATION

It would be useful to have a better sense of the location of the study in order to understand to what degree we can make generalizations about this study. Does this study take place in Denmark? Where precisely? Can we find out more about the population?

SAMPLE

The authors state that the present RCT is limited to those that gave consent. What percentage of participants in this program gave consent to participate? How representative is this group of the target population?

The authors state that there are three groups of participants: “lonely” children, children in foster care, and vulnerable children. How is this determination made? How were participants identified generally? - particularly for the “lonely children”? What does this mean practically? In what kind of family situation are these children living? Again, this information would provide context and offer information regarding the external validity of this study.

INTERVENTION

In terms of the RCT itself, how do we know that the subjects under study themselves responded to this incentive? (And not the parents or caregivers of the children etc.)? - Especially for younger participants (8-12 or so). Also, how long was the questionnaire/ roughly how long did it take? This information could be provided on lines 144-155.

METHODS

The authors state that they randomized the treatment using various procedures in SAS, but the treatment and control groups are quite different in size (146 versus 121). Why is this? Was the randomization successful? Further, in Table 1, it is customary to also provide statistical tests of means post-randomization to verify that the randomization worked correctly (for each control variable). Can this be added? Also, do the authors have any more demographics available to provide additional evidence?

The authors state that “incentives can affect the response rate distribution indirectly by affecting the demographic composition of the study sample or by directly affecting the attitude of the respondents.” While the authors test for differences in survey responses across treatment and control group respondents, they don’t offer an extensive look at the demographics of responders and non-responders in each group (apart from gender and age). Do the authors have any more demographics they could use to investigate to what extent the intervention could have affected the demographic composition of the study sample.

RESULTS

Were those that responded representative of the target population overall? (See notes on demographics above).

It would be helpful to provide not only the response rate and gender percentages but also the raw numbers of individuals (stratified by gender) that participated in the study and responded to the incentives. This way, the reader can clearly see that while these incentives may work better on men, they still do increase the response rate for women as well (in addition to the RR results). For example, these are back-of-the envelope calculations:

Treatment Group

96 women

47 Men

Treated Responders

69 women (72% of women in the treatment group responded)

39 men

Control Group

82 women

37 men

Control Responders

41 women (50% of women in the control group responded)

10 men

Overall, the paper would be stronger if the authors make a better case for why these findings can truly be generalized elsewhere. How similar is the participant population to populations elsewhere? And across the participants (lonely children, foster children, and vulnerable children), are there differences in response rates? As “lonely” is not defined, an analysis of foster and vulnerable children alone could be a useful addition.

6. PLOS authors have the option to publish the peer review history of their article (what does this mean?). If published, this will include your full peer review and any attached files.

Reviewer #1: No

Reviewer #2: No

---

## [Author Response · Author response to Decision Letter 0]

6 Mar 2020

The response to reviewer and editor comments is uploaded in the file Response to reviewers

---

## [Editor Report · Decision Letter 1]

28 Apr 2020

The effect of monetary incentive on survey response for vulnerable children and youths: A randomized controlled trial

PONE-D-19-22509R1

Dear Dr. Pejtersen,

We are pleased to inform you that your manuscript has been judged scientifically suitable for publication and will be formally accepted for publication once it complies with all outstanding technical requirements.

With kind regards,

Dr. Leica S. Claydon-Mueller

Academic Editor

PLOS ONE

---

## [Editor Report · Acceptance letter]

1 May 2020

PONE-D-19-22509R1 

The effect of monetary incentive on survey response for vulnerable children and youths: A randomized controlled trial 

Dear Dr. Pejtersen:

I am pleased to inform you that your manuscript has been deemed suitable for publication in PLOS ONE. Congratulations! Your manuscript is now with our production department. 

With kind regards,

on behalf of

Dr. Leica S. Claydon-Mueller 

Academic Editor

PLOS ONE